# Intraosseous and Intra-Articular Platelet-Rich Plasma for Severe Knee Osteoarthritis: A Real-World-Outcomes Initiative

**DOI:** 10.3390/jcm14113627

**Published:** 2025-05-22

**Authors:** José Miguel Catalán, Gabriel Escarrer-Garau, Maria del Mar Estrany-Celià, Catalina Parra, Laura Arbona-González, Josep Mercader-Barceló, Severiano Dos-Anjos

**Affiliations:** 1Regynere, 07011 Palma, Spain; drcatalan2.0@gmail.com (J.M.C.);; 2Molecular Biology, Health Geography, and One Health Research Group (MolONE), University of the Balearic Islands (UIB), 07122 Palma, Spainsevedos@yahoo.es (S.D.-A.); 3Foners Medicina Veterinària i Innovació SLP, 07006 Palma, Spain; 4VidaCord, 28840 Madrid, Spain

**Keywords:** platelet-rich plasma, knee osteoarthritis, intraosseous injection, real-world clinical data, leukocytes

## Abstract

**Background/Objectives:** The inconsistent clinical outcomes of platelet-rich plasma (PRP) therapy for knee osteoarthritis (KOA) highlight the need to elucidate PRP’s therapeutic potential and influencing factors. Real-world evidence can provide valuable insights in a broad patient population. This study aims to analyze real-world data from KOA patients treated with PRP. **Methods:** Real-world data from 86 KOA patients treated with intraosseous combined with intra-articular (IO + IA) PRP injections were utilized to retrospectively evaluate the long-term effectiveness of this procedure and the impact of PRP characteristics and patient variables on treatment response. **Results:** The WOMAC score was reduced 2 months after completing the treatment procedure. Such a reduction was sustained throughout the follow-up time points, up to 18 months. The percentage of responders was between 55 and 67%. The PRP erythrocyte and leukocyte counts negatively correlated with the change in WOMAC score (ΔWOMAC). The PRP platelet count did not correlate with the WOMAC or the ΔWOMAC scores. The KL severity degree did not affect the responsiveness to treatment. Women reported a higher WOMAC score than men, both before and 2 and 3 months after PRP treatment. However, the ΔWOMAC score was not different between sexes within this period. A negative correlation trend was detected between the patient’s age and the ΔWOMAC score. **Conclusions:** IO + IA PRP administration alleviated severe KOA symptoms with sustained relief for up to 18 months in most patients, potentially delaying the need for knee prostheses. The clinical efficacy was not influenced by narrow platelet dose variations but was negatively impacted by the leukocyte content in the long term, while sex and KOA severity had no influence.

## 1. Introduction

Osteoarthritis is a degenerative joint disease characterized by the progressive degradation of articular cartilage, subchondral bone changes, and synovial inflammation [1]. As a leading cause of disability worldwide, knee osteoarthritis (KOA) particularly imposes a significant burden on healthcare global systems and adversely affects the quality of life of millions of individuals [2]. Osteoarthritis is a leading cause of disability globally, with substantial economic implications and an increasing prevalence due to aging populations [3].

Traditional conservative treatments for KOA, such as nonsteroidal anti-inflammatory drugs (NSAIDs), corticosteroid injections, and physical therapy, often provide only symptomatic relief and fail to halt the progression of the disease. Surgical interventions for severe KOA cases, such as total joint arthroplasty, can provide relief but come with significant risks, including infection, blood clots, and prosthesis-related issues like wear and loosening. Additionally, these procedures require lengthy recovery and extensive rehabilitation. This underscores the need for less invasive treatment options with lower risk profiles [4].

In recent years, platelet-rich plasma (PRP) therapy has emerged as a promising biological treatment for KOA. PRP is an autologous biologic product made of concentrated platelets in plasma, which, when injected into the affected joint, releases a variety of growth factors and cytokines that may facilitate tissue repair and modulate inflammation [5]. These molecules, such as PDGF, IGF-1 or TGF-beta, can modulate inflammation, stimulate chondrocyte proliferation, promote extracellular matrix synthesis, and potentially contribute to cartilage repair and subchondral bone remodeling [1,2,5]. Moreover, PRP exerts immunomodulatory effects by downregulating pro-inflammatory cytokines like IL-1β and TNF-α and inhibiting NF-κB signaling pathways [6].

The potential benefits of PRP include reduced pain, improved joint function, and possibly delayed disease progression, especially for patients affected by mild or moderate KOA. These advantages have led to a growing interest in PRP as a therapeutic option for KOA. Importantly, the therapeutic landscape has also evolved toward combining PRP with surgical techniques, such as microfracture. Some studies suggest that PRP can enhance cartilage regeneration when used in conjunction with microfracture by providing a bioactive environment that supports mesenchymal stem cell recruitment and matrix remodeling [7]. A recent meta-analysis concluded that microfracture combined with PRP leads to significantly better functional scores and cartilage repair outcomes than microfracture alone [8].

Despite the encouraging preclinical findings and positive results from some clinical trials [9,10], the application of PRP in OA treatment has limitations. Variability in PRP preparation methods, differences in platelet concentration and dose, and the lack of standardized treatment protocols contribute to inconsistent clinical outcomes. Furthermore, the exact mechanisms by which PRP exerts its effects are not fully understood, necessitating further research to elucidate its therapeutic potential [11].

A notable gap in the current literature is the scarcity of clinical studies with real-world data. Most existing studies are conducted under controlled conditions with carefully selected patient populations, which may not fully represent the diversity of patients encountered in everyday clinical practice. Real-world data and associated evidence, despite their inherent limitations, can provide valuable insights into the effectiveness and safety of PRP in a broader, more heterogeneous patient population. Such data are crucial for understanding the true clinical utility of PRP and for guiding its integration into standard treatment paradigms for KOA.

While PRP therapy holds promise for the treatment of KOA, especially for moderate disease stages, there is a pressing need for more comprehensive studies that include real-world data. Addressing this need will enhance our understanding of PRP’s role in KOA management and help to establish future evidence-based guidelines for its use.

In this study, utilizing real-world data, we include patients with more severe grades of knee OA to assess long-term clinical outcomes using standardized scales in a real clinical private practice scenario. We analyze the relationship between patient evolution and variables such as KOA severity, sex, age, and platelet dosage. Additionally, each case includes a thorough characterization of the PRP. These findings provide valuable insights into the practical application of PRP in a diverse patient population, reinforcing the importance of real-world data in understanding the efficacy and optimizing the use of PRP in KOA treatment.

## 2. Methods

### 2.1. Study Design

This was a prospective, non-controlled, open clinical study including 86 patients at baseline who presented at a private orthopedic clinic with a diagnosis of KOA of varying severity, enrolled from August 2020 to August 2024. This study was conducted following the guidelines of the Local Ethics Committee, as well as the latest Helsinki Declaration. All the patients provided informed consent to participate in this real-world clinical study, as well as for sample hematological analysis. The patient demographic data are summarized in Table 1.

### 2.2. Inclusion and Exclusion Criteria for Patients

Patients over 18 years of age of any gender with osteoarthritis grade 2–4 according to the Kellgren–Lawrence (KL) radiographic scale were included (Figure 1A). Patients were excluded from the clinical evaluation if they had diabetes mellitus, rheumatoid arthritis, coagulation disorders (coagulopathies), active infections, pregnancy, active tumors, autoimmune diseases, or severe cardiovascular diseases.

### 2.3. Clinical Evaluation

Clinical evaluation was performed using the total WOMAC (Western Ontario and McMaster Universities) scale and the VAS (Visual Analog Scale) comparing with the preoperative condition after 2, 3, 6, 12, and 18 months. The inCytes software (RegenMed LLC, Greenwich, CT, USA) was used for automated follow-up, allowing patients to complete questionnaires at specified intervals using mobile phones, computers, or at the clinic. Only patients with an available baseline evaluation (10 points or more in the WOMAC) and at least one follow-up point were included in this study. Responders were defined as those achieving a ΔWOMAC ≥ 10, based on the minimal clinically important difference established in the previous literature [12].

### 2.4. PRP Obtention and Implantation

PRP was obtained using the Hy-Tissue PRP system from Fidia following the manufacturer’s instructions. Briefly, 50 mL of anticoagulated blood with 3.2% sodium citrate was used to prepare 10 mL of PRP with an average platelet concentration of 550,000/μL. The infiltration protocol described by [13] was slightly modified. Patients received three PRP injections at fifteen-day intervals (Figure 1B). The first treatment involved an intraosseous (IO) infiltration (femoral condyle and tibial plateau, trochlea, or patella based on affected compartments) in the operating room (outpatient setting). IO infiltrations were performed under sedation and local anesthesia using fluoroscopic guidance, manually introducing a 15G trocar into the bone until reaching the subchondral area of the affected spaces. Next, 4 mL of PRP was applied in each IO compartment, with 6–8 mL of PPP (platelet-poor plasma) injected intra-articularly simultaneously. IA injections were preceded by synovial fluid aspiration if detected by ultrasound and PRP activation with calcium chloride. After the injection, the patients were instructed to rest at home and apply ice for pain relief if needed.

### 2.5. PRP Characterization and Hematological Counts

Peripheral blood samples and PRP were obtained from all the patients, and complete blood counts were performed immediately post-extraction. The total number of leukocytes, erythrocytes, and platelets present in the final product was determined using a hematology analyzer (RT-7300, Rayto Life Sciences, Shenzhen, China). See Table 1 for the details of the PRP product characterization applied in the first IO infiltration.

### 2.6. Statistical Analysis

Statistical analyses were carried out using R software (version 4.3.2; R Foundation for Statistical Computing, Vienna, Austria). The normality of the data distribution was tested with the Shapiro–Wilk test. The homogeneity of variance was analyzed with the F or Bartlett test. To compare two variables, paired or independent *t*-tests were applied as appropriate. The correlation of the quantitative variables with the progression of the WOMAC score at the follow-up time points was evaluated using the Spearman correlation coefficient, while the correlation of the qualitative variables was analyzed using the chi-squared test. All the diagrams were created with the R “ggplot2” package, and the graphical data are presented as means with standard error of the mean or displayed in boxplots and dispersion graphs. Subgroup analyses were performed based on treatment responsiveness, PRP platelet concentration, and KL severity degree to analyze the influence of PRP characteristics and patient variables on PRP treatment response.

## 3. Results

### 3.1. Patient and Treatment Characteristics

This study included KOA patients of either sex, without significant differences between the number of both. KOA patients of any age were included. Most of the patients (88%) were diagnosed with a severe KL grade of 3 or 4. The patients’ baseline characteristics are depicted in Table 1. The patients received PRP treatment in which the platelet count was concentrated up to 220% (2.2 times), whereas 65.5% leukocytes and >99.9% erythrocytes were removed (Table 2). A self-reported clinical evaluation was collected 2, 3, 6, 12, or 18 months after PRP treatment and was available, respectively, in 74%, 64%, 59%, 52%, and 42% of the patients recruited at baseline.

### 3.2. Evaluation of the Long-Term Clinical Response to PRP Treatment in KOA Patients

The PRP treatment significantly reduced self-reported pain, stiffness, and functional limitations, as indicated by the significant decline in the WOMAC score, as early as two months after completing the treatment procedure (Figure 2A). Importantly, such a reduction was significantly sustained throughout the follow-up of the patients, up to 18 months after completing the treatment procedure (*p* < 0.001, at each time point).

We assessed the number of patients who reported a significant clinical improvement, termed responders, as defined by a change in the WOMAC score (ΔWOMAC score) of at least 10 points [12] (Figure 2B). The percentage of responders was 55%, 60%, 63%, 67%, and 58% of the total of the patients at each time point, indicating the long-term sustainability of this PRP treatment procedure in a relevant percentage of KOA patients. Moreover, the ΔWOMAC score at 12 months was significantly higher than that presented at 3 months (12.6 vs. 10.9, *p* < 0.05), suggesting that the response might even be magnified across time.

### 3.3. Influence of PRP Characteristics on the Treatment Response in the Long Term

We analyzed the potential influence of variables concerning the PRP characterization (counts of platelets, leukocytes, and erythrocytes) on the treatment responsiveness, disease symptomatology, and the evolution of symptoms in the long term. For the first purpose, the mean values of cell counts were compared between responders and non-responders. For the remaining two purposes, the WOMAC score and ΔWOMAC score were taken as independent variables for correlation analysis.

There were no significant differences between the mean counts of PRP leukocytes, erythrocytes, and platelets between responders and non-responders at any time point (Table 3). However, in the correlation analysis, the PRP variables (concentration or dose) correlated with the WOMAC score or its change. In the whole cohort, negative trends between the PRP leukocyte count and the ΔWOMAC were detected, which reached statistical significance at the 6-month time point. (Figure 3A). Further, in a subgroup analysis in which only responders were included, such negative trends were stronger than in the whole cohort (Figure 3A,B). Within responders, the PRP leukocyte count was inversely and significantly correlated with the ΔWOMAC score at the 6-month time point as well as at the 12-month time point (Figure 3B), suggesting an involvement of the PRP leukocyte concentration in the long-term evolution of symptoms. Regarding erythrocytes, significant negative correlations were detected between the erythrocyte concentration and the WOMAC score at 6 months (*p* < 0.05) or the ΔWOMAC score at 12 months (*p* < 0.05) in responders. Finally, no significant correlations were found between the PRP platelet concentration or dose and the ΔWOMAC score.

Although there was no significant correlation between the injected intraosseous platelet dose and the WOMAC scores or its changes, we further analyzed the potential involvement of the platelet dose on the treatment outcomes because it is discussed as a critical factor for PRP efficacy [14,15]. The KOA patients were split according to the median value of the platelet dose received in the IO administration (4.84 × 10^9^ platelets/injection) into those receiving a high vs. low dose. No significant differences were obtained between patients who received a high (6.0 ± 1.3 × 10^9^ platelets/injection) vs. low (3.8 ± 0.9 × 10^9^ platelets/injection) dose in the WOMAC score or its change nor in the relative frequency between responders and non-responders (Table 4).

### 3.4. Influence of KOA Patient Variables on the PRP Treatment Response in the Long Term

Next, we analyzed whether the patient variables could influence the IO + IA PRP treatment response. The variables analyzed were KOA severity degree at baseline, sex, and age. There were no significant differences in the WOMAC score (Figure 4A) or its change (Figure 4B) between patients with a KL degree of 2–3 and those with a KL degree of 4 at any time point. However, patients with a KL degree of 4 tended to have higher changes in the WOMAC score. Likewise, the KL degree did not affect the responsiveness of the PRP treatment since there were no significant differences in the relative frequency of responders or non-responders depending on their KL degree at any time point (Table 5).

Women reported a significantly higher WOMAC score than men before IO + IA PRP treatment. Women also reported a significantly higher WOMAC score after PRP treatment, which was statistically significant at 2 and 3 months (*p* < 0.005 and *p* < 0.01, respectively), indicating sex-specific differences in the reporting of symptoms (Figure 5A). Despite such differences, sex did not influence the ΔWOMAC score during the first year, but, at 18 months, it was significantly lower in men than in women (*p* < 0.05) (Figure 5B) Although not significant, the relative frequency of male responders was the highest at 2, 3, 6 and 12 months, but this trend was inverted at the last time point (Table 5). Altogether, these data suggest a sex-specific self-reported response of KOA patients to PRP treatment.

The age of the responders was not different from that of the non-responders at any time point (Table 3). Age did not correlate with the WOMAC score or its change when all the patients were considered (*p* > 0.05 for all time-points. Within responders, a negative trend resulted between the patient age and the change in ΔWOMAC after 6 months (*p* = 0.06).

## 4. Discussion

The findings of this study underscore the potential of IO + IA PRP as a promising treatment for severe KOA, providing valuable insights into its clinical efficacy and long-term sustainability. Our results demonstrate significant improvements in patient-reported outcomes, including reductions in pain, stiffness, and functional limitations, as assessed by the WOMAC score, which persisted for up to 18 months post-treatment. Such prolonged benefits emphasize the potential of intraosseous PRP to serve as an effective, minimally invasive therapeutic option for severe KOA, complementing existing conservative and surgical interventions.

It has been proposed that the combined administration of IA + IO PRP has a superior curative potential than the IA PRP injections alone. Compared with isolated IA injections, combined IA + IO PRP administration was more effective in relieving pain, slowing cartilage degeneration, and inhibiting abnormal vascularization and remodeling in a KOA rat model [16]. This hypothesis was supported in recent human studies, involving severe KOA patients [17], and in a controlled clinical trial in which only the patients who received the combined PRP administration reported significant reductions in the synovial-effusion and infra-patellar bursitis at a relatively short term (12 weeks) [9]. However, evidence supporting the efficacy of this combined treatment is still scarce, and additional studies are required to identify the factors that affect KOA patient’s response. In the present study, we investigated the response of KOA patients to the IA + IO PRP administration at long term and analyzed the influence of the PRP- and patient-related factors on the patient’s response by using real-world data.

Most published studies only report clinical follow-ups up to 6 or 12 months [18], and we, here, provide evidence supporting the long-term efficacy of IO + IA PRP injections at platelet dosages of around 5 billion, which is in line with the systematic review by Berrigan and collaborators who reported that those studies having positive outcomes had an average platelet dose of 5.46 billion. We provide valuable information about the sustainability of clinical benefits in a real-world setting. Nevertheless, the optimal dosage of platelets for the treatment of KOA is still a matter of debate [19]. Interestingly, no statistically significant association was found between the platelet concentration (or platelet dose) and the clinical outcomes, a result that might contrast with prior research emphasizing the critical role of platelet dose in PRP efficacy [15,20]. Previous studies have demonstrated that higher PRP concentrations (more than 1 million per microliter, two times more than our mean concentration) correlate with better clinical outcomes, such as reduced pain and improved functional scores (KOOS, IKDC) at 2, 6, and 12 months post-treatment [21]. On the other hand, excessively high platelet concentrations have been shown to inhibit cellular proliferation in vitro, underscoring the importance of optimizing the PRP dose for clinical applications [22]. However, in this study, all the patients received a similar and relatively homogeneous platelet dose (ranging between 3500 and 6500 million in more than 70% of the cases). The average platelet dose in our study was 4800 million platelets (or 4.8 billion), which could be considered as a high dose [15]. Other studies consider 10 billion platelets as the threshold for obtaining better clinical outcomes in KOA intra-articularly, but, in our case, the first injection was intraosseous, and this factor could be affecting the clinical outcome and maybe also the platelet dose to achieve therapeutic efficacy.

The concentration of leukocytes in PRP may also play a pivotal role in predicting clinical outcomes in KOA patients. Elevated levels of TGF-β1, observed in non-responders, have been implicated in fibrosis and synovitis, potentially attenuating the therapeutic benefits of PRP by attracting leukocytes and inducing osteophyte formation [23]. One of our key findings was the observed negative correlation between the PRP leukocyte concentrations and the long-term improvement in WOMAC scores among responders, contrasting with a very recent study [24]. Such discrepancies may be due to the fact that the patients in that study only received IA injections; calcium gluconate as used for activation, and the leukocyte-poor PRP group still had a significant concentration of leukocytes to be a good control. Our data align with prior studies suggesting that higher leukocyte content in PRP may exacerbate inflammation, thereby reducing the therapeutic efficacy of PRP in KOA management [11,25]. Accordingly, leukodepletion (over 88% of total leukocytes) has been considered a determinant factor for the efficacy of PRP for treating KOA [26].

The absence of a significant relationship between the baseline KOA severity (as measured by the Kellgren–Lawrence grade) and the treatment outcomes reinforces the versatility of PRP across different stages of the disease. This finding corroborates previous reports suggesting that PRP may benefit patients with varying degrees of cartilage degeneration [27]. It is important to point out that this dataset includes mostly KL grades 3–4.

Sex-specific differences in the treatment response were another noteworthy finding. Women consistently reported higher WOMAC scores, indicating greater symptom severity both pre- and post-treatment. However, the magnitude of improvement (ΔWOMAC) was comparable between sexes, suggesting that PRP efficacy is not inherently influenced by sex but may interact with self-perception and reporting of symptoms. Our results agree with those of others [28,29], who also reported worse baseline symptomatology in women than in men and comparable final follow-up outcomes between sexes. This observation aligns with the broader literature on sex differences in pain perception and response to musculoskeletal treatments [30].

Patient age might be a key factor in determining the efficacy of PRP treatment in KOA patients. It has been shown that PRP from aged patients contains factors that may suppress chondrocyte matrix synthesis and promote macrophage inflammation in vitro, thereby reducing its therapeutic potential [31]. Indeed, younger patients are more likely to respond to IA PRP injections [32,33], which are in line with the negative trend between the ΔWOMAC at 6 months and patient age observed in our study. Since most of the evidence comes from patients younger than 60, more studies in patients older than 60 are needed to elucidate the impact of KOA patient’s age on the efficacy of PRP treatment.

The use of real-world data in this study provides a broader perspective on the clinical utility of PRP for the treatment of KOA. By including a diverse patient population and employing a non-controlled study design, our findings offer insights into the effectiveness of PRP in everyday clinical practice. Real-world clinical data, while valuable for capturing diverse patient populations and reflecting routine clinical practice, are inherently heterogeneous. However, this approach also introduces inherent limitations, such as variability in patient adherence to follow-up, potential biases in self-reported outcomes, or lack of a control group.

The lack of a placebo or comparative control group is a key limitation, potentially affecting the interpretation of clinical benefit due to placebo effects or disease fluctuation. Moreover, while this study analyses different factors such as hematological characterization of PRP, sex, age, and KOA severity, it lacks data on other potential confounding variables, such as comorbidities (obesity, metabolic syndrome, etc.), habits such as smoking, or the influence of exercise before and/or after treatment. In addition, we lacked data on BMI, comorbidities, and physical activity level, which could be relevant modifiers of the treatment outcome. These unaccounted factors could affect the observed outcomes and should be considered in future studies to better understand their potential impact on PRP treatment efficacy.

## 5. Conclusions

These results obtained from real-world data demonstrate that the combined IO + IA PRP administration reduced KOA symptoms in 67% of the patients affected by severe KOA after 12 months. Further, KOA symptom relief was maintained 18 months after completing the treatment, delaying or avoiding knee prosthesis in the mid-term. Differences in the intraosseous platelet dose, within a narrow range of a mean value of 4.84 × 10^9^ platelets per injection, did not impact clinical efficacy, but the leukocyte content negatively affected the clinical outcome in the long term. Sex and KOA severity degree did not influence the response to this PRP treatment. Our results may contribute to delineating a more optimal PRP treatment procedure for KOA. Nevertheless, these findings derive from a non-controlled, real-world study setting and should be interpreted within the context of its observational design.

## Figures and Tables

**Figure 1 jcm-14-03627-f001:**
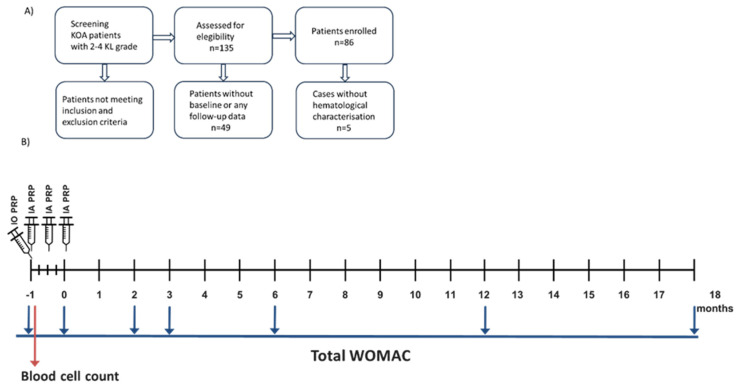
Patient recruitment and study protocol. (**A**) Flowchart illustrates the patient recruitment process. (**B**) Scheme illustrating the clinical study protocol.

**Figure 2 jcm-14-03627-f002:**
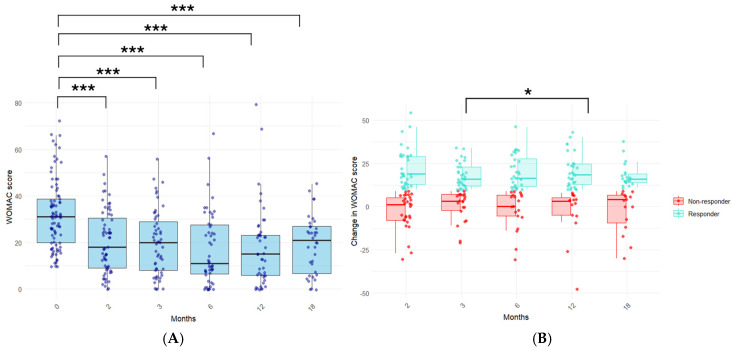
(**A**) Total WOMAC score evolution in knee OA patients. Self-reported pain, joint stiffness, and disability in patients before and at the indicated time points after completing the PRP treatment procedure. (**B**) Change in WOMAC score. The difference in the WOMAC score between baseline and the indicated time points was calculated for each patient. The percentage of responders (those with a ΔWOMAC score of at least 10 points) was 55%, 60%, 63%, 67%, and 58% at 2, 3, 6, 12, and 18 months, respectively. Statistical significance was assessed with paired *t*-tests. *: *p*-value < 0.05; ***: *p*-value < 0.001.

**Figure 3 jcm-14-03627-f003:**
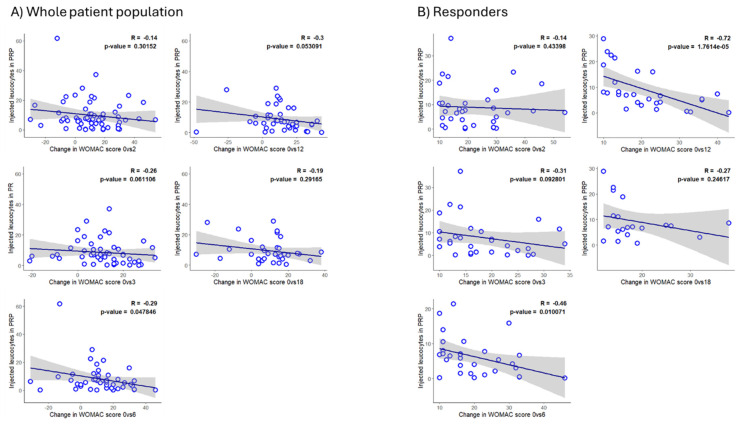
Correlation analysis between PRP leukocyte count and the evolution of OA symptoms. Leukocyte counts in the PRP were confronted with the change in WOMAC score at the indicated time points in the whole cohort (**A**) and in responders (**B**). The Spearman test was used.

**Figure 4 jcm-14-03627-f004:**
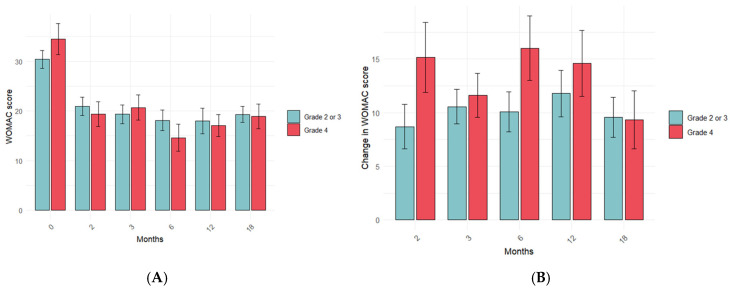
Influence of OA severity on PRP treatment response. The impact of KL severity degree on WOMAC score (**A**) and its change (**B**) was assessed at the indicated time point before and after intraosseous PRP treatment by comparing the mean values. Data are represented as mean values +/− standard error of the mean. The statistical significance was assessed by an independent *t*-test.

**Figure 5 jcm-14-03627-f005:**
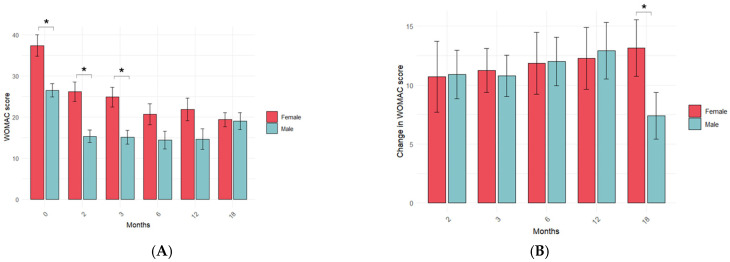
Influence of OA patient sex on PRP treatment response. The impact of sex on WOMAC score (**A**) and its change (**B**) was assessed at the indicated time point before and after intraosseous PRP treatment by comparing the mean values. Data are represented as mean values ± standard error of the mean. The statistical significance was assessed by an independent *t*-test. *: *p*-value < 0.05.

**Table 1 jcm-14-03627-t001:** Demographic data of KOA patients treated with intraosseous PRP. N-miss indicates the number of cases with non-available data. SD: standard deviation.

Sex	
N-Miss	1
Women	42 (49%)
Men	43 (51%)
Age	
Mean (SD)	56.6 (12.9)
Range	15–85
KL severity grade	
Grade 2	10 (11.6%)
Grade 3	46 (53.5%)
Grade 4	30 (34.8%)

**Table 2 jcm-14-03627-t002:** Counts per mL of cells (in millions) present in blood and PRP. N-miss indicates the number of cases with non-available data. SD: standard deviation.

	Blood	PRP
Platelets		
N-miss	5	5
Mean (SD)	226.4 (61.5)	497 (140.4)
Range	94–479	117–850
Leukocytes		
N-miss	4	4
Mean (SD)	5.56 (1.46)	1.92 (5.57)
Range	2.03–10.0	0.01–6.17
Erythrocytes		
N-miss	4	4
Mean (SD)	3865 (780)	0.015 (0.009)
Range	3680–5730	0.00–0.06

**Table 3 jcm-14-03627-t003:** Quantitative variables depending on PRP treatment responsiveness. The mean of patient and treatment variables was calculated in responders and non-responders at each time point after intraosseous PRP administration. Significant differences were assessed with independent *t*-tests.

	Months	Responders	Non-Responders	*p*-Value
WOMAC score	2	14.9	27	<0.001
	3	16.2	25.3	<0.05
	6	10.6	27.8	<0.001
	12	12.3	28.5	<0.005
	18	14.8	25.3	<0.05
Change in WOMAC score	2	22.1	−2.8	<0.001
	3	18	0.4	<0.001
	6	20.3	2.5	<0.001
	12	20.5	−3.1	<0.001
	18	18	−2.5	<0.001
PRP leukocytes (×10^6^/mL)	2	2.12	2.56	0.66
	3	3.37	0.96	0.21
	6	2.87	1.78	0.24
	12	3.62	0.85	0.96
	18	2.42	1.68	0.79
PRP erythrocytes (×10^6^/mL)	2	0.02	0.01	0.60
	3	0.01	0.02	0.48
	6	0.01	0.02	0.067
	12	0.02	0.01	0.30
	18	0.02	0.01	0.87
PRP platelets (×10^6^/mL)	2	502.29	497.86	0.90
	3	481.87	482.29	0.99
	6	487.74	458.76	0.57
	12	486.28	508.21	0.97
	18	488.09	533.57	0.59
Age	2	57.4	55.3	0.24
	3	55.5	60.2	0.34
	6	54.9	59.2	0.47
	12	53.3	58.6	0.63
	18	56.5	54.7	0.31

**Table 4 jcm-14-03627-t004:** Influence of the platelet dose on PRP treatment response in OA patients. The effect of the number of platelets injected in the intraosseous PRP administration on the long-term response to treatment was evaluated by comparing WOMAC score, change in WOMAC score, and the relative frequency of responders between patients receiving a high vs. low platelet dose. The statistical significance was assessed by *t*-test and chi-square test.

	Time (Months)	Low Platelet Dose (3.8 ± 0.9 × 10^9^ Platelets/Injection)		High Platelet Dose (6.0 ± 1.3 × 10^9^ Platelets/Injection)		*p*-Value
WOMAC score	0	31.2		33.5		0.23
	2	21.4		21.3		0.85
	3	19.4		21.7		0.55
	6	16.0		18.7		0.78
	12	16.2		19.4		0.57
	18	17.2		21.7		0.29
Change in WOMAC score	2	9.6		12.0		0.56
	3	10.3		11.8		0.89
	6	12.3		12.1		0.98
	12	12.8		12.9		0.90
	18	10.4		8.7		0.87
Responders		Yes	No	Yes	No	*p*-value
Relative frequencies (%)	2	25.8	24.2	29.0	21.0	0.61
	3	30.8	23.1	28.8	17.3	0.69
	6	31.3	20.8	33.3	14.6	0.49
	12	30.2	18.6	37.2	14.0	0.45
	18	31.4	17.1	28.6	22.9	0.58

**Table 5 jcm-14-03627-t005:** Influence of OA severity on PRP treatment response. The impact of KL severity degree on WOMAC score (A) and its change (B) was assessed at the indicated time point (month) after intraosseous PRP treatment by analyzing the relative frequency of responders. The statistical significance was assessed by the chi-square test.

(A)						
KL Severity Degree		2–3		4		*p*-Value
Responders		Yes	No	Yes	No	
Relative frequencies (%)	2	34.4	32.8	20.3	12.5	0.42
	3	40	23.6	20	16.4	0.57
	6	41.2	27.5	21.6	9.8	0.55
	12	44.4	24.4	22.2	8.9	0.65
	18	38.9	27.8	19.4	13.9	0.99
(B)						
Sex		Women		Men		*p*-value
Responders		Yes	No	Yes	No	
Relative frequencies (%)	2	25	21.9	29.7	23.4	0.84
	3	27.8	20.4	33.3	18.5	0.62
	6	27.4	13.7	35.3	23.5	0.63
	12	22.2	20	44.4	13.3	0.08
	18	30.6	5.6	27.8	36.1	<0.01

## Data Availability

The data that support the findings of this study are available on request from the corresponding author. The data are not publicly available due to privacy or ethical restrictions.

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
