# Peer review of "Intraosseous and Intra-Articular Platelet-Rich Plasma for Severe Knee Osteoarthritis: A Real-World-Outcomes Initiative"

_jcm, 2025, doi:10.3390/jcm14113627_

Round 1

Reviewer 1 Report

Comments and Suggestions for Authors It is interesting the analysis reported on real world data of the efficacy of intraosseus and intra articular platelet richplasmafoe the management of kneeosteoarthritis. It seems that neither prp concentration, nor sex or age affected the outcomes on knee OA. This gives new insigths into the management of kneee OA. In line 227 you referred that women have poorer WOMAC scores before treatment, but even in that case, gender differences did not affect PRP outcomes. It comes in line with Fernandez-Cuadros et. al, who states that females have poorer physyical functioning than men in knee OA, but that does not affect final results. You could cite Fernández-Cuadros, M., Perez-Moro, O. S., Alonso-Sardon, M., Iglesias de Sena, H., & Miron-Canelo, J. A. (2017). Age and sex affect osteoarthritis and the outcome on knee replacement. MOJ Orthop Rheumatol, 9(3), 00091. The follow up of 18 monts is very important since moat studies report follow ups of 6 to 12 months. Figures 3 and 4 do not make referenceto the acronym SEM. Please define acronym SEM in parenthesis at least in figure3.

Author Response

We thank the reviewers for their thorough and constructive feedback. We appreciate the insightful suggestions, which have helped us improve the quality and clarity of the manuscript. Below, we provide a point-by-point response to each of the reviewer’s comments. All relevant changes have also been implemented in the revised manuscript.

It is interesting the analysis reported on real world data of the efficacy of intraosseus and intra articular platelet richplasmafoe the management of kneeosteoarthritis. It seems that neither prp concentration, nor sex or age affected the outcomes on knee OA. This gives new insigths into the management of kneee OA. In line 227 you referred that women have poorer WOMAC scores before treatment, but even in that case, gender differences did not affect PRP outcomes. It comes in line with Fernandez-Cuadros et. al, who states that females have poorer physyical functioning than men in knee OA, but that does not affect final results. You could cite Fernández-Cuadros, M., Perez-Moro, O. S., Alonso-Sardon, M., Iglesias de Sena, H., & Miron-Canelo, J. A. (2017). Age and sex affect osteoarthritis and the outcome on knee replacement. MOJ Orthop Rheumatol, 9(3), 00091. The follow up of 18 monts is very important since moat studies report follow ups of 6 to 12 months. Figures 3 and 4 do not make referenceto the acronym SEM. Please define acronym SEM in parenthesis at least in figure3.

  • Comment 1:

It is interesting the analysis reported on real world data of the efficacy of intraosseous and intra-articular platelet-rich plasma for the management of knee osteoarthritis. It seems that neither PRP concentration, nor sex or age affected the outcomes on knee OA. This gives new insights into the management of knee OA.
In line 227 you referred that women have poorer WOMAC scores before treatment, but even in that case, gender differences did not affect PRP outcomes. It comes in line with Fernandez-Cuadros et al. You could cite:
Fernández-Cuadros, M., Pérez-Moro, O. S., Alonso-Sardón, M., Iglesias de Sena, H., & Mirón-Canelo, J. A. (2017). Age and sex affect osteoarthritis and the outcome on knee replacement. MOJ Orthop Rheumatol, 9(3), 00091.

Response:
We thank the reviewer for their positive appraisal and valuable suggestion. We agree that our findings align with those of Fernández-Cuadros et al., and we have now incorporated this citation in the Discussion section when addressing the influence of sex on WOMAC scores and PRP outcomes (line 291). The added reference reinforces the evidence that, although women report poorer baseline physical functioning, treatment outcomes may remain comparable across sexes.

  • Comment 2:

The follow-up of 18 months is very important since most studies report follow-ups of 6 to 12 months.

Response:
We appreciate this comment. Indeed, one of the strengths of our study is the extended 18-month follow-up period, which provides valuable information about the sustainability of clinical benefits in a real-world setting. We have emphasized this point more clearly in the revised Discussion.

  • Comment 3:

Figures 3 and 4 do not make reference to the acronym SEM. Please define acronym SEM in parenthesis at least in figure 3.

Response:
Thank you for pointing this out. We have now defined the acronym SEM (Standard Error of the Mean) in the caption of the figures (4 and 5 in the revised manuscript), and ensured clarity and consistency across all figure captions.

Reviewer 2 Report

Comments and Suggestions for Authors

A nice study to address Intraosseous and intra-articular PRP for severe knee OA as A Real-World study.

  1. what exactly 'N-Miss' meaning in the table?
  2. A flow chart would helpful for understanding the study design.
  3.  It would be clearer if the authors indicate the exact number of responder in the figures.
  4. In the conclusion,  'in a relevant percentage of patients affected by severe KOA', please add the preside numbers of percentage.

Author Response

We thank the reviewers for their thorough and constructive feedback. We appreciate the insightful suggestions, which have helped us improve the quality and clarity of the manuscript. Below, we provide a point-by-point response to each of the reviewer’s comments. All relevant changes have also been implemented in the revised manuscript.

A nice study to address Intraosseous and intra-articular PRP for severe knee OA as A Real-World study.

  1. what exactly 'N-Miss' meaning in the table?
  2. A flow chart would helpful for understanding the study design.
  3.  It would be clearer if the authors indicate the exact number of responder in the figures.
  4. In the conclusion,  'in a relevant percentage of patients affected by severe KOA', please add the preside numbers of percentage.

Comment 1: the term “N-miss” indicates the number of cases with non-available data for a given parameter. The explanation has been added in tables 1 and 2.

Comment 2: a new figure has been added (Figure 1), which illustrates the recruitment process in a flow-chart (Figure 1A) and the study protocol (Figure 1B).

Comment 3: the number of responders has been indicated in Figure 2B according to the reviewer’s comment.

Comment 4: the precise percentage of responders has been indicated in the conclusions section

Reviewer 3 Report

Comments and Suggestions for Authors

General characteristics and evaluation of the reviewed scientific article:

This article addresses an important clinical topic related to the treatment of severe knee osteoarthritis (KOA) using a combination of intra-articular and intra-injectable PRP (Platelet-Rich Plasma) therapy. The authors present real-world data that are a valuable addition to the randomized controlled trials to date. The paper includes well-described methods, statistical analyses, and interpretation of results.

The paper is interesting, written generally correctly, its structure corresponds to the typical structure of scientific papers, but the article requires major corrections and additions to both content and references before further processing and acceptance for publication. Below are my detailed comments and observations.

Minor comments:

Expanding the discussion of osteoarthritis in the introduction could substantially strengthen this section by emphasizing the clinical, societal, and economic relevance of this highly prevalent condition. Osteoarthritis is not only a leading cause of pain and disability worldwide but also a growing public health concern due to increasing life expectancy and lifestyle-related risk factors. Its prevalence is shaped by a multifactorial interplay of determinants, including occupational demands, high-impact or repetitive sports activities, prior musculoskeletal injuries, obesity, and biological sex differences. A comprehensive overview of these risk factors—critically examined and supported by recent literature—would provide a strong contextual foundation for the topic and underscore the need for continued research. The following references are particularly recommended for inclusion in this section: https://doi.org/10.3390/healthcare12161648 and DOI: 10.1056/NEJMcp1903768.

Please expand the description of PRP in the introduction and add the latest references. I recommend adding: Short-term effects of arthroscopic microfracturation of knee chondral defects in osteoarthritis;  Microfracture combined with platelet rich plasma for cartilage injury: A meta analysis; Platelet-rich plasma combined with microfracture versus microfracture in the treatment of knee cartilage lesions: A meta-analysis;

The lack of comparison with a placebo group or a group with another treatment significantly limits the interpretation of the effectiveness of the therapy. The authors should clearly emphasize this weakness in the discussion and present the possible consequences of the lack of control (placebo effect, regression to the mean, natural history effect of the disease). Please significantly expand the description of limitations.

The article does not include information on factors such as body weight, BMI, comorbidities (e.g. obesity, metabolic syndrome), or physical activity, which may affect the results of treatment. Future studies should take into account and analyze the influence of these variables. In the current text, it is worth adding a note about these deficiencies in the study limitations.

The criterion of response to treatment (ΔWOMAC ≥ 10 points) was adopted without clear justification and reference to the established MCID (minimal clinically important difference) values. The choice of the threshold should be clearly justified and appropriate literature sources that confirm this threshold should be cited.

The effect of age on treatment effects is presented only as a "trend" without in-depth analysis. It is worth conducting an additional multivariate analysis with age as a covariate to confirm the independence of this effect. Alternatively, the interpretation of the results in the context of the literature is more detailed.

The text and tables do not always clearly indicate whether the data are statistically significant; the phrase "n.s." (not significant) is often used without a p value. I recommend adding specific p values ​​for all analyses, even if they are not significant - this will increase clarity and allow for a more precise interpretation.

The conclusions are formulated in a general way and do not emphasize the methodological limitations. The "Conclusions" section should clearly state that the results concern a population with a high burden of disease and were obtained in real clinical practice, without control and randomization.

Congratulations on an interesting study and I wish you continued scientific success.

Author Response

We thank the reviewers for their thorough and constructive feedback. We appreciate the insightful suggestions, which have helped us improve the quality and clarity of the manuscript. Below, we provide a point-by-point response to each of the reviewer’s comments. All relevant changes have also been implemented in the revised manuscript.

General characteristics and evaluation of the reviewed scientific article:

This article addresses an important clinical topic related to the treatment of severe knee osteoarthritis (KOA) using a combination of intra-articular and intra-injectable PRP (Platelet-Rich Plasma) therapy. The authors present real-world data that are a valuable addition to the randomized controlled trials to date. The paper includes well-described methods, statistical analyses, and interpretation of results.

The paper is interesting, written generally correctly, its structure corresponds to the typical structure of scientific papers, but the article requires major corrections and additions to both content and references before further processing and acceptance for publication. Below are my detailed comments and observations.

Minor comments:

Expanding the discussion of osteoarthritis in the introduction could substantially strengthen this section by emphasizing the clinical, societal, and economic relevance of this highly prevalent condition. Osteoarthritis is not only a leading cause of pain and disability worldwide but also a growing public health concern due to increasing life expectancy and lifestyle-related risk factors. Its prevalence is shaped by a multifactorial interplay of determinants, including occupational demands, high-impact or repetitive sports activities, prior musculoskeletal injuries, obesity, and biological sex differences. A comprehensive overview of these risk factors—critically examined and supported by recent literature—would provide a strong contextual foundation for the topic and underscore the need for continued research. The following references are particularly recommended for inclusion in this section: https://doi.org/10.3390/healthcare12161648 and DOI: 10.1056/NEJMcp1903768.

Please expand the description of PRP in the introduction and add the latest references. I recommend adding: Short-term effects of arthroscopic microfracturation of knee chondral defects in osteoarthritis;  Microfracture combined with platelet rich plasma for cartilage injury: A meta analysis; Platelet-rich plasma combined with microfracture versus microfracture in the treatment of knee cartilage lesions: A meta-analysis;

The lack of comparison with a placebo group or a group with another treatment significantly limits the interpretation of the effectiveness of the therapy. The authors should clearly emphasize this weakness in the discussion and present the possible consequences of the lack of control (placebo effect, regression to the mean, natural history effect of the disease). Please significantly expand the description of limitations.

The article does not include information on factors such as body weight, BMI, comorbidities (e.g. obesity, metabolic syndrome), or physical activity, which may affect the results of treatment. Future studies should take into account and analyze the influence of these variables. In the current text, it is worth adding a note about these deficiencies in the study limitations.

The criterion of response to treatment (ΔWOMAC ≥ 10 points) was adopted without clear justification and reference to the established MCID (minimal clinically important difference) values. The choice of the threshold should be clearly justified and appropriate literature sources that confirm this threshold should be cited.

The effect of age on treatment effects is presented only as a "trend" without in-depth analysis. It is worth conducting an additional multivariate analysis with age as a covariate to confirm the independence of this effect. Alternatively, the interpretation of the results in the context of the literature is more detailed.

The text and tables do not always clearly indicate whether the data are statistically significant; the phrase "n.s." (not significant) is often used without a p value. I recommend adding specific p values ​​for all analyses, even if they are not significant - this will increase clarity and allow for a more precise interpretation.

The conclusions are formulated in a general way and do not emphasize the methodological limitations. The "Conclusions" section should clearly state that the results concern a population with a high burden of disease and were obtained in real clinical practice, without control and randomization.

Congratulations on an interesting study and I wish you continued scientific success.

  • Comment 1: Expanding the discussion of osteoarthritis in the introduction could substantially strengthen this section. The reviewer recommends including references: https://doi.org/10.3390/healthcare12161648 and DOI: 10.1056/NEJMcp1903768.

Response: We agree with the reviewer and have expanded the introduction to provide a more comprehensive overview of the clinical, societal, and economic relevance of knee osteoarthritis. We have incorporated the recommended references to reinforce the context and need for new therapeutic options like PRP.

  • Comment 2: Please expand the description of PRP in the introduction and add the latest references.

Response: As suggested, we have expanded the introduction to provide a more detailed description of PRP, its mechanisms of action, and classification. We have also included additional relevant references, including recent meta-analyses and clinical data supporting intraosseous and intra-articular PRP combinations.

  • Comment 3: The lack of comparison with a placebo group or another treatment group limits the interpretation of effectiveness. Please emphasize this in the discussion.

Response: We acknowledge this important limitation. It was already included in the discussion section, but we have added a specific paragraph in the Discussion section addressing the lack of a control group, highlighting potential confounding factors such as placebo effects, regression to the mean, and natural disease progression. This limitation is now clearly acknowledged and its impact on interpretation is discussed.

  • Comment 4: Add a note in the limitations about the lack of information on BMI, comorbidities, and physical activity.

Response: We appreciate this suggestion, but all those limitations are already included in the last paragraph of the discussion section. We have specifically included a new sentence in this regard.

  • Comment 5: The criterion of response to treatment (ΔWOMAC ≥ 10 points) lacks justification. Please provide reference to established MCID values.

Response: We thank the reviewer for pointing this out. We have now justified the use of a ΔWOMAC ≥ 10 as a threshold by citing relevant literature supporting its validity as a minimal clinically important difference in patients with KOA. It was already included in the Results section, but now also appears in Methods (clinical evaluation)

  • Comment 6: The effect of age is presented as a trend only. Consider multivariate analysis or expand interpretation.

Response: As suggested by the reviewer, we performed a multivariate analysis including both patient and PRP variables to further analyze the potential influence of age in KOA progression and treatment response. The multivariate analysis did not further add new information on the effect of patient age, and therefore our conclusions remain the same.

Comment 7: The phrase 'n.s.' is used in tables without specific p-values. Add p-values even if not significant.

Response: We have revised all tables to include specific p-values for each comparison, even when not statistically significant. We agree that this improves transparency and interpretability.

Comment 8: Conclusions are general and do not mention methodological limitations. Emphasize this clearly.

Response: We have revised the conclusions to clearly state that the study was performed in a real-world, non-controlled setting with inherent methodological limitations. The revised section now reflects a more cautious interpretation of findings and their generalizability.

  • Comment 9: Congratulations on an interesting study and I wish you continued scientific success.

Response: We sincerely thank the reviewer for their encouraging and thoughtful comments.